# ADAMS project: a genetic Association study in individuals from Diverse Ancestral backgrounds with Multiple Sclerosis based in the UK

Benjamin M Jacobs ![ORCID],[1] Luisa Schalk,[1] Angie Dunne,[2] Antonio Scalfari,[3] Ashwini Nandoskar,[4] Bruno Gran,[5] Charles A Mein,[6] Charlotte Sellers,[1] Cord Spilker,[7] David Rog,[8] Elisa Visentin,[9] Elizabeth Lindsey Bezzina,[10] Emeka Uzochukwu,[11] Emma Tallantyre,[11,12] Eva Wozniak,[6] Eve Sacre,[2] Ghaniah Hassan-Smith,[13] Helen L Ford,[2] Jade Harris,[14] Joan Bradley,[15] Joshua Breedon,[1] Judith Brooke,[14] Karim L Kreft,[16] Katherine Tuite Dalton,[17] Katila George,[1] Maria Papachatzaki,[18] Martin O'Malley,[2] Michelle Peter,[1] Miriam Mattoscio,[19] Neisha Rhule,[20] Nikos Evangelou,[21] Nimisha Vinod,[14] Outi Quinn,[7] Ramya Shamji,[9] Rashmi Kaimal,[1] Rebecca Boulton,[21] Riffat Tanveer,[22] Rod Middleton,[17] Roxanne Murray,[1] Ruth Bellfield,[7] Sadid Hoque,[1] Shakeelah Patel,[22] Sonia Raj,[22] Stephanie Gumus,[18] Stephanie Mitchell,[14] Stephen Sawcer,[23] Tarunya Arun,[24] Tatiana Pogreban,[9] Terri-Louise Brown,[1] Thamanna Begum,[1] Veronica Antoine,[14] Waqar Rashid,[25] Alastair J Noyce,[1] Eli Silber,[10] Huw Morris,[26] Gavin Giovannoni,[1] Ruth Dobson ![ORCID] [1]

For numbered affiliations see end of article.

**Correspondence to**
Dr Ruth Dobson;
ruth.dobson@qmul.ac.uk

## ABSTRACT

**Purpose** Genetic studies of multiple sclerosis (MS) susceptibility and severity have focused on populations of European ancestry. Studying MS genetics in other ancestral groups is necessary to determine the generalisability of these findings. The genetic Association study in individuals from Diverse Ancestral backgrounds with Multiple Sclerosis (ADAMS) project aims to gather genetic and phenotypic data on a large cohort of ancestrally-diverse individuals with MS living in the UK.

**Participants** Adults with self-reported MS from diverse ancestral backgrounds. Recruitment is via clinical sites, online (https://app.mantal.co.uk/adams) or the UK MS Register. We are collecting demographic and phenotypic data using a baseline questionnaire and subsequent healthcare record linkage. We are collecting DNA from participants using saliva kits (Oragene-600) and genotyping using the Illumina Global Screening Array V.3.

**Findings to date** As of 3 January 2023, we have recruited 682 participants (n=446 online, n=55 via sites, n=181 via the UK MS Register). Of this initial cohort, 71.2% of participants are female, with a median age of 44.9 years at recruitment. Over 60% of the cohort are non-white British, with 23.5% identifying as Asian or Asian British, 16.2% as Black, African, Caribbean or Black British and 20.9% identifying as having mixed or other backgrounds. The median age at first symptom is 28 years, and median age at diagnosis is 32 years. 76.8% have relapsing–remitting MS, and 13.5% have secondary progressive MS.

**Future plans** Recruitment will continue over the next 10 years. Genotyping and genetic data quality control are

## STRENGTHS AND LIMITATIONS OF THIS STUDY

⇒ The ADAMS project will be a large, ancestrally diverse, genotyped cohort of individuals with multiple sclerosis (MS), which will help to extend genetic analysis of MS to people from non-European ancestral backgrounds. Online recruitment facilitates rapid scaling of the study, with minimal imposition on participants. The current recruitment rate demonstrates the feasibility of reaching our recruitment targets.

⇒ Online recruitment is likely to introduce various biases, for instance, it may discourage older and more disabled individuals from signing up.

⇒ Relying on online self-report for MS diagnosis and phenotype information may not be as accurate as clinician-determined measures—there is a small risk that people without clinically definite MS may sign up for the study.

⇒ This is a case-only cohort. The accuracy of case–control analyses relies on the abundance of ancestrally similar controls in UK Biobank and other control datasets.

⇒ Much larger sample sizes are required to make novel genetic discoveries.

ongoing. Within the next 3 years, we aim to perform initial genetic analyses of susceptibility and severity with a view to replicating the findings from European-ancestry studies. In the long term, genetic data will be combined with other datasets to further cross-ancestry genetic discoveries.

## INTRODUCTION

The genetic architecture of multiple sclerosis (MS) susceptibility in individuals of European ancestry has been extensively assessed.[1–5] Case–control genome wide association studies (GWAS) performed by the International Multiple Sclerosis Genetics Consortium (IMSGC) have discovered 233 independent signals across the genome strongly associated with MS risk, collectively explaining ~39% of heritability.[2] The strongest signals lie within the major histocompatibility complex locus on chromosome 6, with the DRB1*15:01 allele conferring the largest effect of any allele (OR: ~3).[6] More recently, genetic analysis has also shed light on the genetic drivers of disease severity.[7]

To date, genome-wide screening efforts to identify determinants of MS risk in populations of non-European ancestry have only involved modest numbers of participants, thus providing limited statistical power.[8] Unsurprisingly, none of these studies has identified new genome-wide significant associations, however the single nucleotide polymorphisms (SNPs) associated with MS susceptibility in genetically European populations have shown concordant effects on risk in populations of South Asian, African, Hispanic and ancestrally mixed backgrounds.[9–16] It is inevitable that many of the variants of relevance in MS will differ in allele frequency between populations. This will potentially result in differences in power to identify such variants in specific populations, emphasising the value of studying the genetic architecture of MS in diverse cohorts.[16 17]

Patterns of linkage disequilibrium (LD, the correlation between genetic variants) also differ greatly between ancestral populations. Due to LD, it can be challenging to identify the underlying causal variant/s responsible for a region of genetic association. Populations with less extensive LD—such as populations of African ancestry—afford greater power for fine-mapping as the number of variants which could plausibly account for any given GWAS signal tends to be lower.[8 18] The complementary patterns of LD between ancestries can also be leveraged to further improve fine-mapping.

Genetic studies of diverse populations are also expected to improve the generalisability of downstream post-GWAS applications such as polygenic score prediction and Mendelian randomisation.[8] In addition to the scientific advances expected from this avenue of research, broadening participation in medical research is valuable both in itself and for its instrumental societal impact.[19]

Here we report the design and initial cohort phenotype results of a UK-based genetic study of MS risk and severity in individuals of diverse ancestral backgrounds (a genetic Association study in individuals from Diverse Ancestral backgrounds with Multiple Sclerosis [ADAMS]). We are prospectively recruiting individuals with MS from diverse ancestries via a web-based platform and via clinical routes with a view to performing genetic analysis of MS susceptibility and severity. The long-term goal of this project is to combine these data with international datasets to facilitate multi-ancestry genetic analysis of MS.

## COHORT DESCRIPTION

### Recruitment

The ADAMS project is an ongoing genetic cohort study of individuals with MS from diverse ancestral backgrounds living in the UK (https://app.mantal.co.uk/adams). We are recruiting individuals with self-reported MS via a bespoke online platform, clinical sites and the UK MS Register (UKMSR).[20 21] We are working with networks of primary care practices across the UK where patients have consented for contact regarding research to ensure wide reach of this study. Individuals with a coded diagnosis of MS on their primary care records are contacted with the study information and directed to the study website. Individuals who have previously signed up for the UKMSR can consent to participate in ADAMS from their UKMSR 'home page' at any point. In addition, the UKMSR study team sends regular emails to participants when they are due to complete new online participant-reported outcome measures. These emails contain information about new studies, including ADAMS. Individuals can also sign up via one of our 15 participating clinical sites (full list of participating sites is given in the online supplemental file 1). Finally, individuals who hear about the study via social media or other public engagement channels can self-refer via the website.

### Inclusion and exclusion criteria

This study is focused on people with MS from diverse ancestral backgrounds, that is, people with recent non-European ancestry. However, self-reported ethnicity or ancestry is not a strict inclusion/exclusion criterion. Our rationale for this is that our study aims to be as inclusive as possible, and self-reported ethnicity is a relatively crude and poor proxy for genetic ancestry. We are, therefore, recruiting from a diverse audience and will infer genetic ancestry from genotyping data as part of the data analysis pipeline.

Inclusion and exclusion criteria are summarised below.

#### Inclusion criteria

► Self-reported MS (diagnosis will be validated against clinical records for a subset of participants)
► Willing and able to give informed consent
► Age >18 years.
► Willing and able to provide a saliva sample
► Willing to answer baseline survey questions.

#### Exclusion criteria

► Unable to consent.
► Already participating in another study from which we are going to use genomic data (UK Biobank [UKB] and Genes and Health).

### Data collection and genotyping

Once participants have provided informed consent via the dedicated study website, they are directed to a baseline questionnaire. A set of core data elements are collected, including basic demographic details, details about their MS, an address for dispatching the saliva kit and their

NHS number for linkage to their medical records. When participants have completed the demographic questionnaire, they are sent an Oragene-600 saliva kit along with a pre-paid envelope to return to our laboratory. In addition to these data, we are in the process of administering validated questionnaires via our platform to formally assess participants' Expanded Disability Scale Score (EDSS), Multiple Sclerosis Impact Scale 29 and the EuroQol EQ-5D-5L, which assesses overall health-related quality of life. The current consent form and participant information sheet can be downloaded from https://app.mantal.co.uk/adams/consent?hl=GB.

The website (https://app.mantal.co.uk/adams) was co-designed with persons with MS. All new data generated as part of this project are stored securely on Queen Mary University of London (QMUL) servers. Phenotype and genotype data are stored separately and linked by pseudonymous IDs.

Participants are asked to provide consent for researchers to access their medical notes, and for their details to be used to link to national data systems (i.e. via NHS Digital). For the initial phase of the study, a subset of participants recruited through the primary NHS site (Barts Health NHS Trust) will have their medical records checked by clinically qualified investigators who will collect a set of core data elements to establish the accuracy of self-reported information submitted by participants. The core data elements set will include: age at symptom onset, age at diagnosis, initial MRI date and findings, initial cerebrospinal fluid findings, including the presence/absence of oligoclonal bands, initial diagnostic criteria applied, clinical diagnosis (including MS subtype), history of disease-modifying treatment use, EDSS at onset, list of dates and manifestations of clinical relapses, and use of walking aids.

Genotyping is being performed in batches using the Illumina Global Screening Array V.3 with multi-disease booster content (GSA v3+EAMD) in collaboration with the Genome Centre at QMUL.[15 22 23]

### Patient and public involvement
Individuals with MS from a range of different ethnic backgrounds were involved in this study from its inception and have ongoing input into the management, design and communications from the study. We are working with a group of participants who help to run the study.

### External datasets and control datasets
We will combine data with large external control datasets for case–control genetic analysis. We will be using data from Genes and Health, a genetic study of ~50 000 individuals of British South Asian ancestry,[23] and UKB.[24] UKB contains genetic data for ~9000 individuals of South Asian genetic ancestry and ~7000 of African genetic ancestry.[25] Both datasets contain details of significant medical diagnoses of participants, meaning that control-only populations can be derived. These high quality pre-existing genetic datasets will be used as controls

for case–control analysis. Additional genetic data from individuals with MS from diverse genetic backgrounds genotyped as part of other studies will be obtained via collaborators.

### Target recruitment and power calculations
We have performed power calculations to assess the expected number of cases required to replicate European MS susceptibility alleles (i.e. find association at a nominal, unadjusted p value of <0.05). We calculated power for each of the 233 European MS susceptibility SNPs by assuming an equivalent marginal effect size across ancestries, taking allele frequency differences between ancestries into account (https://github.com/benjacobs123456/popPoweR). We used discovery-stage summary statistics from the IMSGC GWAS to determine European-ancestry effect sizes.[2]

These calculations demonstrate that recruiting ~300 cases within each ancestral cluster will yield reasonable power to replicate the European susceptibility alleles. We will have power to detect approximately 193/289 and 176/289 of the European risk alleles in South Asian and African-ancestry individuals, respectively, with a case sample size of n=300. Mean power across all susceptibility variants in the African population is estimated at 61% and at 67% in the South Asian population. With n=500 in each ancestral group, mean power increases to 69% (African ancestry) and 75% (South Asian ancestry). As many of the association signals within the Major Histocompatibility Complex (MHC) are of large magnitude and genome-wide significant effects are in the order of OR 1.2–1.3, we anticipate good power to prove replication at most of the ~233 genome-wide significant loci within this ancestrally diverse cohort. We will not have statistical power to discover novel associations through this cohort with the current recruitment target. Furthermore, it is important to note that these power calculations are rough estimates—and may be overestimates—as we do not yet know the precise ancestral composition of the cohort, and there is significant genetic diversity within these broad ancestral clusters.

### Analytic plan
We will conduct genetic analysis of MS susceptibility and severity within each ancestral cluster. Ancestral groups will be defined using principal component analysis and global ancestry inference. Genome-wide association testing will be performed using mixed logistic or linear models (for case–control and continuous phenotypes, respectively). We will perform a meta-analysis on these results with GWAS summary statistics from European-ancestry populations using random and fixed-effect meta-analysis. A variety of post-GWAS analyses will be conducted, including cross-ancestry genetic correlation, fine mapping, polygenic risk score profiling and Mendelian randomisation.

## FINDINGS TO DATE

Recruitment started in November 2021 and is currently expected to run until August 2031 pending long-term funding. As of 3 January 2023, we have recruited 682 participants (n=446 via the website, n=55 in person at clinical sites and n=181 via the UKMSR). Of this initial cohort, 71.2% of participants are female, with a median age of 44.9 years at recruitment (IQR: 18.3). Over 60% of the cohort are non-white British, with 23.5% identifying as Asian or Asian British, 16.2% as Black, African, Caribbean or Black British and 20.9% identifying as having mixed or other backgrounds. A sizeable proportion (10.7%) prefer to not disclose their ethnic background, and 28.7% identify as white. For the substantial proportion of the cohort born outside the UK (23.1%), the median age at migration was 18 years (IQR: 20 years).

We have gathered self-reported data from participants to facilitate genetic analyses of MS phenotypes. The median age at first symptom is 28 years (IQR: 15 years), and the median age at diagnosis 32 years (IQR: 14 years). 76.8% have relapsing–remitting MS, and 13.5% have secondary progressive MS. The majority (70.3%) are currently receiving and/or have previously received disease-modifying therapy. Forty three per cent of our cohort stated that they are unlimited in their mobility, and 36.8% report using a walking aid at time of recruitment (approximate Expanded Disability Status Scale score of at least 6.0).

Questionnaire data also provides an insight into risk factors for MS. Twenty per cent of participants reported being overweight during adolescence, 16.8% report a family history of MS, 47.2% report ever having smoked and 15.3% report having had glandular fever.

The average age at first reported symptom tended to be earlier in individuals who identified as South Asian (mean: 26.4 years (SD: 8.3)) compared with self-reported Black individuals (mean: 30.6 years (SD: 12.1), p=0.02) and white individuals (mean: 31.6 years (SD: 10.4), p<0.0001). We observed a similar pattern for age at self-reported diagnosis, with black (mean: 32.9 years (SD: 12.1), p=0.04) and Asian (mean: 29.7 years (SD: 8.5), p<0.0001) individuals reporting an earlier age at diagnosis than white individuals (mean: 36.5 years (SD: 10.1)). The gender balance of the recruited cohort also differed between ethnic groups (p=0.03), with a higher proportion of men in the self-reported Asian cohort (39% male) compared with the white (25.6% male) and black cohorts (22.4% male). These differences in demographic characteristics, although based on a small cohort so far, are consistent with previous findings.[26]

## CONCLUSIONS

ADAMS (https://app.mantal.co.uk/adams) is a genetic cohort study aiming to determine the genetic basis of MS risk and severity in individuals from non-European ancestral backgrounds living in the UK. Recruitment and genotyping are ongoing. Individuals with MS or potential collaborators can contact the study team via adams_study@qmul.ac.uk.

**Author affiliations**
[1]Preventive Neurology Unit, Wolfson Institute of Population Health, Queen Mary University of London, London, UK
[2]Leeds Centre for Neurosciences, Leeds teaching Hospitals NHS Trust, Leeds, UK
[3]Centre of Neuroscience, Department of Medicine, Imperial College London, London, UK
[4]Hillingdon and Imperial NHS Trust, London, UK
[5]Department of Neurology, Nottingham University Hospitals NHS Trust, Mental Health and Clinical Neuroscience Academic Unit, University of Nottingham School of Medicine, Nottingham, UK
[6]Barts and the London Genome Centre, Queen Mary University of London, London, UK
[7]Bradford Teaching Hospital Foundation Trust, Bradford, UK
[8]Manchester Centre for Clinical Neurosciences, Northern Care Alliance NHS Trust, Manchester, UK
[9]Research and Innovation, Queen's Hospital, BHRUT, London, UK
[10]Kings College Hospital and Lewisham and Greenwich NHS Trusts, London, UK
[11]Division of Psychological Medicine and Clinical Neurosciences, Cardiff University, Cardiff, UK
[12]Department of Clinical Neurology, University Hospital of Wales, Cardiff, UK
[13]University Hospitals Birmingham NHS Foundation Trust, Birmingham, UK
[14]Northern Care Alliance NHS Trust, Manchester, UK
[15]Hillingdon and Imperial NHS Trust, Uxbridge, UK
[16]Department of Neurology, University Hospital of Wales, Cardiff, UK
[17]Population Data Science, Swansea University Medical School, Swansea, UK
[18]Mid and South Essex NHS Foundation Trust, Southend-on-Sea, UK
[19]Department of neuroscience, Queen's Hospital, BHRUT NHS Trust, Romford, UK
[20]Queen Elizabeth Hospital (Lewisham and Greenwich NHS Trust), London, UK
[21]Department of Neurology, Nottingham University Hospitals NHS Trust; Mental Health and Clinical Neuroscience Academic Unit, University of Nottingham School of Medicine, Nottingham, UK
[22]Lancashire Teaching Hospital NHS Foundation Trust, Preston, UK
[23]University of Cambridge, Department of Clinical Neuroscience, Addenbrookes Hospital, Hills Road, Cambridge, UK
[24]University Hospitals of Coventry and Warwickshire, Coventry, UK
[25]St George's University Hospitals NHS Foundation Trust, London, UK
[26]Department of Clinical and Movement Neuroscience, UCL Queen Square Institute of Neurology, London, UK

**Acknowledgements** A full list of contributing authors and their contributions can be found in the supplement. BJ, RD, HRM, GG, ES and AJN were involved in the conception of the study. BJ and RD have ultimate oversight over the study. BJ and LS are responsible for the day-to-day management of the study. All authors are involved in data acquisition and/or recruitment. All authors were involved in drafting and editing the manuscript, have given final approval for submission, and accept accountability for the work reported.

**Collaborators** Alastair J Noyce, Angie Dunne, Antonio Scalfari, Benjamin M Jacobs, Bruno Gran, Charles A Mein, Charlotte Sellers, Cord Spilker, David Rog, Eli Silber, Elisa Visentin, Elizabeth Lindsey Bezzina, Emeka Uzochukwu, Emma Tallantyre, Eva Wozniak, Eve Sacre, Gavin Giovannoni, Helen L Ford, Huw Morris, Jade Harris, Joshua Breedon, Judith Brooke, Karim L Kreft, Katila George, Luisa Schalk, Martin O'Malley, Michelle Peter, Miriam Mattoscio, Neisha Rhule, Nimisha Vinod, Outi Quinn, Ramya Shamji, Rashmi Kaimal, Rod Middleton, Roxanne Murray, Ruth Bellfield, Ruth Dobson, Sadid Hoque, Stephanie Mitchell, Stephen Sawcer, Tarunya Arun, Tatiana Pogreban, Terri-Louise Brown, Thamanna Begum and Veronica Antoine.

**Contributors** BJ and RD designed the study and have ultimate oversight over the study. BJ wrote the first draft of the manuscript, is responsible for the day-to-day management of the study and analysed the data. RD is responsible for the overall content as the guarantor.

**Funding** This study is funded by an Medical Research Council (MRC) Clinical Research Training Fellowship (CRTF) jointly supported by the UK MS Society (BMJ; grant reference: MR/V028766/1), AIMS2CURE (grant reference: N/A) and Barts Charity. This work is being carried out at the Preventive Neurology Unit at Queen Mary University of London, which is partly funded by Barts Charity.

**Competing interests** None declared.

**Patient and public involvement** Patients and/or the public were involved in the design, or conduct, or reporting, or dissemination plans of this research. Refer to the Methods section for further details.

**Patient consent for publication** Not applicable.

**Ethics approval** This study involves human participants. This study and its amendments have been approved by the London - South East Research Ethics Committee (reference: 21/PR/1289). Participants gave informed consent to participate in the study before taking part.

**Provenance and peer review** Not commissioned; externally peer reviewed.

**Data availability statement** No data are available. Genetic association summary statistics will be made publicly available on completion of the pre-planned analyses. Genetic data for participants who sign up via the UK MS Register will be fed back into their secure data safe haven and will be available to bona fide researchers on request at https://ukmsregister.org/. Pseudonymised individual-level genotype data will be made available via the European Genome-Phenome Archive. We are seeking collaborators for data sharing and/or widening our recruitment network. Please contact the lead author or the generic study email address (adams_study@qmul. ac.uk) for further information.

**ORCID iDs**
Benjamin M Jacobs http://orcid.org/0000-0002-6023-6010
Ruth Dobson http://orcid.org/0000-0002-2993-585X

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
