## [Reviewer comments · BMJ Open]

ARTICLE DETAILS

TITLE (PROVISIONAL)	Cohort profile: The ADAMS project - a genetic Association study in individuals from Diverse Ancestral backgrounds with Multiple Sclerosis based in the United Kingdom
AUTHORS	Jacobs, Benjamin; Investigators, ADAMS; Dobson, Ruth

VERSION 1 – REVIEW

REVIEWER	Okai, Annette University of North Texas Institute for Healthy Aging
REVIEW RETURNED	25-Jan-2023

GENERAL COMMENTS	This project will be a significant contribution to the field of MS and provide valuable insight into the changing demographics of the MS population. Minor revision; recognizing the paucity of data on this topic, there are a few more recent paper on genetic susceptibility and would recommend including those as well.
--

REVIEWER	Hrastelj , James Cardiff University, Division of Psychological Medicine and Clinical Neuroscience
REVIEW RETURNED	16-Feb-2023

GENERAL COMMENTS	Summary: The manuscript is essentially a protocol for the creation of a long term cohort. It describes the rationale for, methodology of and initial recruitment to a long term cohort of patients with multiple sclerosis from diverse ancestries. Demographic, clinical and genetic data will be collected. Methods: Recruitment to the study is via an online portal, via clinical sites or by the MS registry. MS diagnosis is self-reported, but with validation in a sub-group. Demographic data for the cohort suggests it is successfully recruiting patients from diverse ethnic backgrounds. The clinical data collected so far relating to multiple sclerosis is consistent with other cohorts. In order to identify novel genetic risk variants in subpopulations, the sample size needs to be large. Although there are power calculations included, there is the possibility that the size of this cohort would not be large enough to provide sufficient power. However, the authors mitigate against this by planning to combine this cohort with other cohorts internationally. They are collecting DNA via saliva. Previous work in our lab has shown the quality and quantity of the DNA obtained from saliva to be less than blood. This failure rate should be accounted for. They plan to use the Illumina Global Screening Array v3 chip with multi-disease booster. It will be important to ensure this chip's coverage is adequate in the genetic populations recruited. Understanding the
---

	genetic aetiology of disease severity could be an exciting avenue for identifying modifiable targets, however recent severity studies have suggested that the genetic basis for severity is limited. This will be important to explore, but may not be a very productive analysis. Furthermore, because the cohort we will not be population based it will likely to be incompletely representative of the MS population at large. This will introduce significant bias to the severity analysis. Importance/impact: There is a huge bias in genetic and epidemiological studies towards Caucasian European populations. There is an urgent need to address this. It is likely that there are genetic risk variants for multiple sclerosis that are specific to certain ethnic groups. Furthermore, genetic risk variants that have already been identified need to be replicated in other genetic populations in order to better understand the aetiology of multiple sclerosis in different ethnicities and allow results to be more generalisable. These issues are of critical importance for better understanding the aetiology and clinical course of the disease. The authors are very well placed to create numerous publications from this study that will reference this manuscript.
--	--

REVIEWER	MacDougall, Niall Hairmyres Hospital, Department of Neurology I am involved in the FutureMS studies in Scotland which involve genetic information and multiple sclerosis. I am also involved in the Scottish MS Register.
REVIEW RETURNED	02-Mar-2023

GENERAL COMMENTS	Good paper outlining an important study that ultimately should add a lot to the medical literature and our understanding of the genetics of MS. It would be useful to clearly state when recruitment started and when it is expected to end. Initial recruitment seems to have been good. Good discussion of limitations of study. Given self-reported nature of participants misdiagnosis must be a potential limitation here too.
--

REVIEWER	Kearns, Patrick The University of Edinburgh Centre for Clinical Brain Sciences, Anne Rowling Regenerative Neurology Clinic
REVIEW RETURNED	09-Mar-2023

GENERAL COMMENTS	This is an impressive study that reports preliminary data collected from 682 participants from under-studied ancestral backgrounds who are living with multiple sclerosis. The study is timely, and ongoing, with plans for an ambitious 10 year recruitment. This description of the cohort will be of interest to researchers in the wider field of population genetics. Reporting the study whilst recruitment is ongoing is sensible as this may attract collaborators and potentially participants and/or study sites. That MS genetic studies have been performed predominantly in participants of European ancestry is without doubt (Genetics for all. Nat Genet 51, 579 (2019). https://doi.org/10.1038/s41588-019-0394-y). The authors should be congratulated on their efforts to involve multiple study sites and facilitate engagement and recruitment from under-represented groups.
---

	I have no major concerns with the manuscript. Some comments follow: -- The 10-year recruitment plan is ambitious. Is funding secured for this? If so, this is a major strength that should be made explicit given that long-term resourcing is a significant challenge for research of this kind. -- Abstract: "Studying MS genetics in other ancestral groups is necessary to ensure the generalisability of these findings." I would suggest that non-generalisability of specific findings (e.g. SNP-associations) from European ancestry studies to other populations would be one of the most interesting discoveries that could be made with this study, as it would suggest that the genetic context or environmental context of the modified the causal effect. I would suggest ensuring generalisability would be an anti-climax. and would suggest reframing. -- "Adults with self-reported Multiple Sclerosis from diverse (i.e. non-European) ancestral backgrounds." I understand that these individuals are self-reported to be of non-European ancestry. Although, the inclusion and exclusion criteria don't make this clear how this is defined. If people of European ancestry register, are they then excluded. At the risk of being pedantic, I would also suggest changing to "non-European recent ancestry", and attempting to define recent, as everyone has non-European ancestry. I think the study would benefit from defining participants' ancestry for inclusion as clearly as possible: how much non-European and how recently? E.g. at least one grand-parent born outside of Europe. This would be useful even if only for the purposes of stratifying participants after they have been recruited. --"In addition to the scientific advances expected from this avenue of research, there is an urgent need for researchers to ensure that trial and study cohorts are diverse and inclusive of the whole population." A discussion of the statistical downsides of broadening participation would be helpful in the interest of balance. I don't disagree with the statement specifically in this case, but there are other perspectives, and there is legitimate debate as to whether trials or cohort studies should even aim for representativeness at all. It depends on the study objectives. See, for example: https://www.fharrell.com/post/ia/ https://pubmed.ncbi.nlm.nih.gov/24062290/ https://www.ncbi.nlm.nih.gov/pmc/articles/PMC3888189/ https://academic.oup.com/ije/article/42/4/1018/658638 -- Will samples from the study be stored for future analyses beyond SNP-genotyping? I presume not given the resources involved, but this will be helpful information for potential collaborators. I would recommend making what will be collected, used, and stored explicit. Both for the collected data variables and for the biological samples, if possible. -- For the African-ancestry subgroup, much greater genetic diversity exists within African populations than elsewhere, do the authors anticipate this affecting their power to replicate associations or to otherwise complicate their analyses? -- The study is recruiting participants in the UK with recent non-
--	---

	European ancestries, suggesting that participants (23.1%) or their recent ancestors have migrated to the UK. How do the authors intend to tackle the issue of migration (and particularly migration age), given compelling literature suggests an environmental contribution to MS risk acts particularly during childhood (https://pubmed.ncbi.nlm.nih.gov/31081503/). Are the authors collecting migration histories? It is plausible that amongst even the individuals born in the UK, a significant proportion may have lived outside the UK for periods prior to recruitment given ancestry ties. -- The earlier age at diagnosis in Asian population is interesting. As is the apparent sex discrepancy. As is the fact that the authors have observed this in an apparently independent study. Have similar findings been previously reported by other authors or in other locations or during other study periods? It would be interesting to know if the authors consider this age discrepancy likely to be the result of earlier diagnosis due to earlier disease onset or increased access to healthcare/awareness or some other reason. Is disability similar across these groups after adjusting for age?
--	--

VERSION 1 – AUTHOR RESPONSE

Reviewer: 1

Dr. Annette Okai, University of North Texas Institute for Healthy Aging

Comments to the Author:

This project will be a significant contribution to the field of MS and provide valuable insight into the changing demographics of the MS population.

Minor revision; recognizing the paucity of data on this topic, there are a few more recent paper on genetic susceptibility and would recommend including those as well

Thank you - we have updated the reference list with two studies published since initial manuscript submission.

Reviewer: 2

Dr. James Hrastelj , Cardiff University

Comments to the Author:

Summary: The manuscript is essentially a protocol for the creation of a long term cohort. It describes the rationale for, methodology of and initial recruitment to a long term cohort of patients with multiple sclerosis from diverse ancestries. Demographic, clinical and genetic data will be collected.

Methods: Recruitment to the study is via an online portal, via clinical sites or by the MS registry. MS diagnosis is self-reported, but with validation in a sub-group. Demographic data for the cohort suggests it is successfully recruiting patients from diverse ethnic backgrounds. The clinical data collected so far relating to multiple sclerosis is consistent with other cohorts. In order to identify novel genetic risk variants in subpopulations, the sample size needs to be large. Although there are power calculations included, there is the possibility that the size of this cohort would not be large enough to provide sufficient power. However, the authors mitigate against this by planning to combine this cohort with other cohorts internationally. They are collecting DNA via saliva. Previous work in our lab has shown the quality and quantity of the DNA obtained from saliva to be less than blood. This failure rate should be accounted for. They plan to use the Illumina Global Screening Array v3 chip with multi-disease booster. It will be important to ensure this chip's coverage is adequate in the genetic populations recruited. Understanding the genetic aetiology of disease severity could be an exciting avenue for identifying modifiable targets, however recent severity studies have suggested that the genetic basis for severity is limited. This will be important to explore, but may not be a very productive analysis. Furthermore, because the cohort we will not be population based it will likely to be incompletely representative of the MS population at large. This will introduce significant bias to the severity analysis.

Thank you - we agree there is likely to be bias in the severity analysis, and have acknowledged this as the top limitation in the bullet points. On the note of saliva, we agree the DNA can be problematic for whole-genome sequencing but our preliminary genotyping data shows a failure rate of <5%, most of which is due to issues with sample leakage/low volumes of saliva. We have also included some references to the Genes & Health study which has successfully used this approach (Oragene saliva sampling followed by GSA genotyping) to genotype ~50,000 people of British South Asian ancestry.

Re power, we agree that this cohort alone will not have power to detect novel associations, but will be valuable a) for replication of European signals and b) as a building block for future cross-ancestry genetic analyses. We have made this explicit in the text, writing “We will not have statistical power to discover novel associations through this cohort.”.

Importance/impact: There is a huge bias in genetic and epidemiological studies towards Caucasian European populations. There is an urgent need to address this. It is likely that there are genetic risk variants for multiple sclerosis that are specific to certain ethnic groups. Furthermore, genetic risk variants that have already been identified need to be replicated in other genetic populations in order to better understand the aetiology of multiple sclerosis in different ethnicities and allow results to be more generalisable. These issues are of critical importance for better understanding the aetiology and clinical course of the disease. The authors are very well placed to create numerous publications from this study that will reference this manuscript.

Reviewer: 3

Dr. Niall MacDougall, Hairmyres Hospital, Institute of Neurological Sciences

Comments to the Author:

Good paper outlining an important study that ultimately should add a lot to the medical literature and our understanding of the genetics of MS.

It would be useful to clearly state when recruitment started and when it is expected to end. Initial recruitment seems to have been good.

We have clarified this in the text.

Good discussion of limitations of study. Given self-reported nature of participants misdiagnosis must be a potential limitation here too.

Thank you - we have made this explicit in the limitations section: “- Relying on online self-report for MS diagnosis and phenotype information may not be as accurate as clinician-determined measures – there is a small risk that people without clinically-definite MS may sign up for the study.”

Reviewer: 4

Dr. Patrick Kearns, The University of Edinburgh Centre for Clinical Brain Sciences, The University of Edinburgh MRC Human Genetics Unit

Comments to the Author:

This is an impressive study that reports preliminary data collected from 682 participants from under-studied ancestral backgrounds who are living with multiple sclerosis. The study is timely, and ongoing, with plans for an ambitious 10 year recruitment. This description of the cohort will be of interest to researchers in the wider field of population genetics. Reporting the study whilst recruitment is ongoing is sensible as this may attract collaborators and potentially participants and/or study sites. That MS genetic studies have been performed predominantly in participants of European ancestry is without doubt (Genetics for all. Nat Genet 51, 579 (2019). <https://doi.org/10.1038/s41588-019-0394-y>). The authors should be congratulated on their efforts to involve multiple study sites and facilitate engagement and recruitment from under-represented groups.

I have no major concerns with the manuscript. Some comments follow:

-- The 10-year recruitment plan is ambitious. Is funding secured for this? If so, this is a major strength that should be made explicit given that long-term resourcing is a significant challenge for research of

this kind.

Thank you - currently we have funding that will support the study until 2024. We have made it clear in the text we will depend on more funding to continue recruitment for the full ethically-approved period.

-- Abstract: "Studying MS genetics in other ancestral groups is necessary to ensure the generalisability of these findings."

I would suggest that non-generalisability of specific findings (e.g. SNP-associations) from European ancestry studies to other populations would be one of the most interesting discoveries that could be made with this study, as it would suggest that the genetic context or environmental context of the modified the causal effect. I would suggest ensuring generalisability would be an anti-climax. and would suggest reframing.

Thank you - we have rephrased this as 'determine' rather than 'ensure'. We have been a bit deliberately cautious with our language because available data from small studies done so far in non-European ancestries have suggested that it is generally the same loci that crop up, and it is likely to be the same causal variants - what is likely to differ will be the tagging effects due to different LD patterns and allele frequencies. Truly ancestry-specific loci may exist, but realistically that will take sample sizes of >10,000 to find which is beyond the scope of our efforts at the moment.

-- "Adults with self-reported Multiple Sclerosis from diverse (i.e. non-European) ancestral backgrounds."

I understand that these individuals are self-reported to be of non-European ancestry. Although, the inclusion and exclusion criteria don't make this clear how this is defined. If people of European ancestry register, are they then excluded. At the risk of being pedantic, I would also suggest changing to "non-European recent ancestry", and attempting to define recent, as everyone has non-European ancestry. I think the study would benefit from defining participants' ancestry for inclusion as clearly as possible: how much non-European and how recently? E.g. at least one grand-parent born outside of Europe. This would be useful even if only for the purposes of stratifying participants after they have been recruited.

Thank you - this is an important point and we have clarified in the text. We are essentially recruiting anyone but with a strong push towards people who are not White British. We will not be using any self-reported ancestry/ethnicity in the analysis as generally these are very poor proxies for genetic ancestry, which we will infer from the genotyping data. We have deliberately not gone into detail on the genetic analysis plan as the degree of granularity which we can use will depend on sample size - at present our plan is to conduct broad analyses within each 1,000 Genomes superpopulation (African, South Asian). Controlling for genetic principal components within these clusters should control most population stratification. The more samples we get, the more granular we can be (e.g. we may eventually have enough power to conduct separate analyses for people of Bangladeshi vs Indian ancestry).

We are not excluding anyone from the study once they sign up. Everyone's data will be useful, the European-ancestry samples will make a relatively smaller contribution to what is known already given the work done by IMSCG already (GWAS of 50,000 cases, 70,000 controls of European ancestry).

In the text we have clarified "This study is focussed on people with MS from diverse ancestral backgrounds, i.e. people with recent non-European ancestry. However, self-reported ethnicity or ancestry is not a strict inclusion/exclusion criterion. Our rationale for this is that our study aims to be as inclusive as possible, and self-reported ethnicity is a crude and poor proxy for genetic ancestry. We are therefore recruiting from a diverse audience and will infer genetic ancestry from genotyping data as part of the data analysis."

--"In addition to the scientific advances expected from this avenue of research, there is an urgent need for researchers to ensure that trial and study cohorts are diverse and inclusive of the whole population."

A discussion of the statistical downsides of broadening participation would be helpful in the interest of balance. I don't disagree with the statement specifically in this case, but there are other perspectives, and there is legitimate debate as to whether trials or cohort studies should even aim for

representativeness at all. It depends on the study objectives.

See, for example:

<https://www.fharrell.com/post/ia/>

<https://pubmed.ncbi.nlm.nih.gov/24062290/>

<https://www.ncbi.nlm.nih.gov/pmc/articles/PMC3888189/>

<https://academic.oup.com/ije/article/42/4/1018/658638>

Thank you - we agree this is an interesting topic and the purely statistical argument is more controversial. We were really pointing to the purely ethical case here rather than the statistical argument - in our view one of the important arguments for doing research in under-served populations is that representation is a good thing both intrinsically and because of the downstream impact it has: it fosters trust in medical professionals and can only improve the relationship between patients and doctors. We have included a short reference to support this, but intentionally mention it only briefly as we are not exploring the ethical case in detail.

-- Will samples from the study be stored for future analyses beyond SNP-genotyping? I presume not given the resources involved, but this will be helpful information for potential collaborators. I would recommend making what will be collected, used, and stored explicit. Both for the collected data variables and for the biological samples, if possible.

We have made this explicit in the text: "Genetic association summary statistics will be made publicly-available on completion of the pre-planned analyses. Genetic data for participants who sign up via the UK MS Register will be fed back into their secure data safe haven and will be available to bona fide researchers on request at <https://ukmsregister.org/>." We will probably also be able to share". Leftover DNA after genotyping is unlikely to be publicly-available as the volumes are relatively small and we will probably end up doing exome sequencing on these samples if we secure more funding.

-- For the African-ancestry subgroup, much greater genetic diversity exists within African populations than elsewhere, do the authors anticipate this affecting their power to replicate associations or to otherwise complicate their analyses?

Yes - this is an important point and we have clarified in the text that these figures will be overestimates in the presence of significant heterogeneity: "These power calculations are rough estimates – and may be overestimates - as we do not yet know the precise ancestral composition of the cohort, and there is significant genetic diversity within these broad ancestral clusters."

-- The study is recruiting participants in the UK with recent non-European ancestries, suggesting that participants (23.1%) or their recent ancestors have migrated to the UK. How do the authors intend to tackle the issue of migration (and particularly migration age), given compelling literature suggests an environmental contribution to MS risk acts particularly during childhood (<https://pubmed.ncbi.nlm.nih.gov/31081503/>). Are the authors collecting migration histories? It is plausible that amongst even the individuals born in the UK, a significant proportion may have lived outside the UK for periods prior to recruitment given ancestry ties.

Thanks for raising this important topic. We hope that migration will not be a major confounder in the study and attempt to collect data on it as far as possible. Although obviously migration will not impact on genetic ancestry, we acknowledge that non-random migration and non-random participation in the study could introduce biases. In the absence of major gene-by-environment effects, e.g. a hypothetical genetic variant that only increases the risk/severity of MS in the presence of some environmental risk factor, it is unlikely that migration will be a major confounder for either the susceptibility or the severity analyses. Note that there is no empirical evidence for a gene-by-environment effect of this magnitude in MS, and there are scant examples across other diseases.

We have collected data on age at migration and country of birth. We have included age in the text.

-- The earlier age at diagnosis in Asian population is interesting. As is the apparent sex discrepancy. As is the fact that the authors have observed this in an apparently independent study. Have similar findings been previously reported by other authors or in other locations or during other study periods? It would be interesting to know if the authors consider this age discrepancy likely to be the result of

earlier diagnosis due to earlier disease onset or increased access to healthcare/awareness or some other reason. Is disability similar across these groups after adjusting for age?

We agree this is interesting - we think the evidence from other cohorts is more compelling due to the biases inherent in our study (self-recruitment, mainly online recruitment). The preprint for our CPRD paper is online at https://papers.ssrn.com/sol3/papers.cfm?abstract_id=4237729. Given the small cohort size so far and the biases we would prefer to not elaborate on this too much just yet. In preliminary analyses, we have found that there are no differences between ethnicities in terms of age-adjusted disability measures (i.e. the ARMSS score), but this is preliminary and not yet ready for inclusion in the manuscript.

VERSION 2 – REVIEW

REVIEWER	MacDougall, Niall Hairmyres Hospital, Department of Neurology I participate in other clinical research projects including the FutureMS cohort study which has a genetic component
REVIEW RETURNED	06-Apr-2023
GENERAL COMMENTS	I note the revisions and response to reviewer comments and I think things have been dealt with well.
REVIEWER	Kearns, Patrick The University of Edinburgh Centre for Clinical Brain Sciences, Anne Rowling Regenerative Neurology Clinic
REVIEW RETURNED	03-Apr-2023
GENERAL COMMENTS	The authors have addressed all the previous comments satisfactorily. The study is likely to make valuable discoveries and I agree with the authors' ethical arguments for broadening participation to under-represented groups and with their conclusion that these trump the statistical debate in this case. Congratulations on an important piece of work.